# Fruiting Body Heterogeneity, Dimorphism and Haustorium-like Structure of *Naematelia aurantialba* (Jin Er Mushroom)

**DOI:** 10.3390/jof10080557

**Published:** 2024-08-07

**Authors:** Ying Yang, Caihong Dong

**Affiliations:** 1State Key Laboratory of Mycology, Institute of Microbiology, Chinese Academy of Sciences, Beijing 100101, China; yangy@im.ac.cn; 2University of Chinese Academy of Sciences, Beijing 100049, China

**Keywords:** *Naematelia aurantialba*, *Stereum hirsutum*, heterogeneity, dimorphism, haustoria

## Abstract

Mushroom Jin Er has attracted widespread attention in Asia over the past two decades due to its medicinal properties and nutritional values. In the present study, Jin Er basidiocarps were often found to be surrounded by *Stereum hirsutum* fruiting bodies in their natural habitat and occasionally in artificial cultivation. The observation of two different kinds of mycelia within the hymenium and analyses of ITS sequences confirmed that Jin Er basidiocarps were composed of two fungal species, *Naematelia aurantialba* and *S. hirsutum*. This heterogeneity of Jin Er fruiting bodies is indeed distinct from the homogeneous hypha of *Tremella fuciformis* found in Yin Er mushroom, although its development also requires the presence of another fungus *Annulohypoxylon stygium*. Basidiospores can germinate on the surface of basidiocarps and produce mycelia. However, basidiospores in PDA medium can only bud into yeast-like conidia. The yeast-like conidia of *N. aurantialba* can transform into pseudohyphae with a change in temperature from 20 °C to 28 °C or switch into filamentous cells on an induction medium (IDM) at 20 °C, 25 °C and 28 °C. This dimorphic was reported for the first time in *N. aurantialba*. Haustorium-like structures were abundantly observed both within the hymenium and in the aerial mycelia cultured on the IDM. The developmental process was documented firstly in this study, involving the formation of protuberances with basal clamp connections, elongation at the protuberances, branch production, and eventual maturation. However, further observation is required to determine whether the haustorium-like structures can penetrate *S. hirsutum* hyphae. These findings are expected to provide valuable insights into the relationship and interaction between these two fungi, thereby advancing the cultivation of fruiting bodies.

## 1. Introduction

*Naematelia aurantialba* (Bandoni & M. Zang) Millanes & Wedin, a jelly fungus, has gained popularity across Asia over the past two decades, being widely utilized as both a tonic food and a medicinal product. It was previously classified as *Tremella aurantialba* [1] until it was scientifically revised to *N. aurantialba* based on molecular data and physiochemical and morphological features [2]. The fruiting bodies of *N. aurantialba* are typically characterized by a gelatinous or rubbery texture and a bright orange-to-yellow color. This species is commonly known as “Jin Er” or “brain Er”, referencing the distinct color and shape of the basidiocarps. 

A substantial quantity of polysaccharides have been identified within *N. aurantialba,* and their potential health benefits have been extensively researched, including hypoglycemic effects [3], antitumor properties [4], and immunity-enhancing effects [5]. This edible mushroom is naturally found in the north temperate regions across several provinces of China, including Yunnan, Sichuan, Fujian, Qinghai, Gansu, and Xizang Autonomous Region [6]. However, wild resources are scarce, leading to its classification as a vulnerable species in the ‘Redlist of China’s Biodiversity −Macrofungi’ by the Ministry of Ecology and Environment of the People’s Republic of China and the Chinese Academy of Sciences in 2018 (https://www.mee.gov.cn/xxgk2018/xxgk/xxgk01/201805/t20180524_629586.html, accessed on 1 August 2024). Cultivated *N. aurantialba* fruiting bodies will become a good substitute for wild resources.

After years of hard work, *N. aurantialba* fruiting bodies have now been successfully cultivated on an industrial scale [7,8,9]. It has been established that the cultivation of Jin er fruiting bodies necessitates the presence of another fungus, *Stereum hirsutum* [10,11]. A particular blend of *N. aurantialba* and *S. hirsutum*, termed the effective spawn, is crucial for successful cultivation. The ratio of these two fungi during spawn preparation and their cultivation conditions are quite important. The utilization of inferior or ineffective spawn may lead to issues such as varying shapes of fruiting bodies from mushroom growing bags, occasional failure to set fruit, and low biological conversion rates. Therefore, securing effective spawn is essential to achieving both high yield and superior quality in artificial cultivation [12,13]. Nonetheless, the relationship and interaction between these two fungi are not fully understood, resulting in a lack of a scientific foundation for the preparation of an effective spawn. A comprehensive grasp of the fundamental biological characteristics and development of the fungus, encompassing morphology, habitat, growth conditions, and reproduction processes, would aid in elucidating the relationship and interaction between these fungi.

In the present study, the macro- and micro-morphologies of *N. aurantialba* basidiocarps were observed in detail. Two reproductive strategies of *N. aurantialba* basidiospores were identified: they either germinated into hyphae, or they budded into yeast-like conidia. The dimorphism between yeast-like conidia and pseudohyphae or hyphae in *N. aurantialba* was reported for the first time. Finally, the morphology and developmental process of the haustorium-like structure of *N. aurantialba* were described. These findings are expected to be valuable in understanding the relationship and interaction between *N. aurantialba* and *S. hirsutum* and advancing fruiting body cultivation.

## 2. Materials and Methods

### 2.1. Fungal Specimen, Strains and Culture Conditions 

Wild Jin Er fruiting bodies were collected from Gyirong (HMAS No. 271220) and Nyingchi (HMAS No. 265185), Xizang Autonomous Region, China. The cultivated fruiting bodies were collected from Beijing (HMAS No. 259471), Yunnan, Hebei and Gansu provinces. The strains of *N. aurantialba* and *S. hirsutum* used in the study were isolated from wild fruiting bodies of Nyingchi, Xizang Autonomous Region, and deposited in the China General Microbiological Culture Collection Center (CGMCC) under numbers CGMCC 5.2267 and CGMCC 5.2266, respectively. To maintain these strains, they were cultured on Potato Dextrose Agar (PDA: 200 g potato, 20 g dextrose, 18 g agar, 1000 mL water) medium at 23 °C for 7 days. Basidiospores were collected by the spore ejection method [14] and kept on a PDA medium. The yeast-like conidia were incubated on PDA medium at 23 °C for 5–7 days and kept at 4 °C. For the study of dimorphism, the yeast-like conidia were incubated on an induction medium (IDM) in addition to PDA medium. The IDM was prepared by boiling 100 g of solid residue remaining after harvesting the *N. aurantialba* fruiting bodies with 1 L of water for 20 min. After filtering the mixture through gauze to remove the solid residue, the resulting filtrate was collected, and 24 g of agar was added.

### 2.2. DNA Extraction and Detection

DNA was extracted from different locations (the apex, middle and base) of both wild and cultivated *N. aurantialba* basidiocarps, basidiospores of *N. aurantialba*, and pure mycelia of *S. hirsutum* using the modified Cetyltrimethylammonium Bromide (CTAB) method [15], and ITS sequences were amplified with ITS1 (5′-TCCGTAGGTGAACCTGCGG-3′)/ITS4 (5′-TCCTCCGCTTATTGATATGC-3′) primers [16]. The PCR amplification system comprised 12.5 μL of Taq 2× Master Mix (No. P222-w1, Vazyme Biotech Co., Ltd., Nanjing, China), 0.5 μL of ITS1 and 0.5 μL of ITS4, 1 μL of genome DNA, and 10.5 μL of distilled water. The PCR amplification reaction procedure involved pre-denaturation at 94 °C for 3 min, followed by 35 cycles of denaturation at 94 °C for 30 s, annealing at 55 °C for 30 s, extension at 72 °C for 15 s, and final extension at 72 °C for 5 min. The amplification products were detected by 1% agarose gel electrophoresis, and distinct bands were cut and purified with the Gel DNA Extraction Kit (GE706-200, Genesand Biotech Co., Ltd., Beijing, China), following the manufacturer’s instructions. Finally, the purified DNA samples were sequenced by Sangon Biotech (Shanghai) Co., Ltd., Shanghai, China.

### 2.3. Phylogenetic Analysis

The ITS sequences of *N. aurantialba* and *S. hirsutum* were blasted against National Center for Biotechnology Information (NCBI) databases, respectively. Several sequences with similarity were downloaded. The nucleotide sequences were aligned using MEGA version 11.0 [17] with default parameters. A maximum likelihood phylogenomic tree was constructed using the conserved nucleotide sequences in MEGA version 11.0. One thousand bootstrap replicates were performed, and the trees were visualized using FigTree v1.4.2 [18]. Bootstrap values higher than 50% were indicated in the phylogenetic trees.

### 2.4. Observation by Microscopy and SEM

Hymenium and haustorium-like structures were observed using an optical microscope (Eclipse 80i, Nikon, Tokyo, Japan) and a scanning electron microscope (SEM) (SU8010, Hitachi, Tokyo, Japan). For the SEM observation, 0.5 cm^2^ of the fruiting body of *N. aurantialba* was fixated with 2.5% glutaraldehyde (McLean Biochemical Technology, Shanghai, China) in 0.1 M buffer at 4 °C for 24 h. The glutaraldehyde was eluted three times with deionized water. Subsequently, the deionized water was dehydrated using a series of ethanol solutions at concentrations of 70%, 85%, 95%, and 100%, respectively. Finally, the sample was glued onto stubs covered with a thin conductive layer and observed by SEM following critical-point drying and gold sputtering [19,20]. The *N. aurantialba* fruiting body was examined under a dissecting microscope (SMZ18, Nikon, Tokyo, Japan) to observe its surface details, particularly the white mycelia mat on the surface of *N. aurantialba* basidiocarps (Jin Er).

## 3. Results

### 3.1. Naematelia aurantialba Basidiocarps Are Composed of Two Fungal Species

*N. aurantialba* basidiocarps are relatively large, with dimensions ranging from 3.5 to 11 cm in thickness and 6.5 to 12 cm in length (Figure 1A,B). The cerebriform shape refers to their appearance, resembling a convoluted brain. In nature, these basidiocarps grow on dead or dying branches of broadleaf trees. Notably, the *N. aurantialba* basidiocarps are often found surrounded by fruiting bodies of the fungus *S. hirsutum* (Figure 1A). Occasionally, the fruiting bodies of *S. hirsutum* were observed on the mushroom growing bag, alongside the cultivation of *N. aurantialba* fruiting bodies, particularly when the environmental conditions were not suitable for *N. aurantialba* growth (Figure 1C). 

To determine whether the mycelia of *S. hirsutum* were included in the *N. aurantialba* basidiocarps, DNA was extracted from various locations of *N. aurantialba* basidiocarps (Figure 1D), and ITS sequences were amplified. Simultaneously, DNA samples from basidiospores of *N. aurantialba* and pure mycelia of *S. hirsutum* were also extracted and amplified. The results showed that the DNA extracted from different locations of *N. aurantialba* basidiocarps yielded two distinct bands (Figure 1E), corresponding to the amplification results of the *N. aurantialba* basidiospores (GenBank accession number PP859876.1) and the pure mycelia of *S. hirsutum* (GenBank accession number PP859881.1), respectively. Following gel purification, sequencing and phylogenetic analysis (Figure 1F,G), the two bands were identified as *N. aurantialba* (GenBank accession number PQ084585.1) and *S. hirsutum* (GenBank accession number PQ084986.1), respectively. Wild Jin Er fruiting bodies collected from Gyirong and Nyingchi in the Xizang Autonomous Region of China, as well as the artificially cultivated Jin Er fruiting bodies, demonstrated the consistent results (Figure 1D and Appendix A). This result confirmed that *N. aurantialba* basidiocarps were composed of two fungal species, *N. aurantialba* and *S. hirsutum*. The coexistence of these mycelia within the basidiocarp creates a unique and distinct structure. In our subsequent research, we will denote *N. aurantialba* basidiocarps “Jin Er” to differentiate them effectively.

### 3.2. Microscopic Structure of Hymenium

The basidia, basidiospores, conidia, and mycelia within the hymenium of Jin Er of HMAS No. 271220 and HMAS No. 265185 were observed. Probasidia, the early stages of basidia formation, were formed by the swollen apex of the hyphae, their initial shape was ellipsoid or olive (Figure 2A), and their size ranged from (5) 5.5–11.5 (12) × (3) 3.5–8.0 (8.5) μm with a *Q*-value (ratio of length to width) of approximately 1.5. The matured basidia were observed as pyriform (pear-shaped), broadly ellipsoid, or subglobose (Figure 2B), and their size was (11.5) 12–23.5 (23) × (12.5) 13–22.0 (22.5) μm, with a *Q*-value of approximately 0.9. The cells were primarily four-celled and exhibited cruciate septation, indicating that they were divided into four compartments by cross-walls (Figure 2C). Epibasidia, which were additional structures, emerged from the four-celled basidia with thin walls. These structures formed sterigmata at their apex (Figure 2D), providing attachment points for basidiospores. The basidiospores were subglobose to globose, broadly ellipsoid, and distinctly blunt apiculate (having a small, pointed tip), with particulate inclusions (Figure 2E). Their size ranged from (7.0) 8.5–12.5 (14.0) × (6.5) 8.0–11.0 (11.5) μm, with a *Q*-value of approximately 1.10. The morphological observations align with the descriptions provided by Bandoni and M. Zang in 1990 [1]. Basidiospores were collected and identified as *N. aurantialba* by ITS sequence analyses (Figure 1F), and there was no contamination.

In addition to basidiospores, the fungus also produced asexual spores. These asexual spores, known as conidia, were formed on scattered conidiophores, typically part of the specialized conidiogenous hyphae, within the hymenium (Figure 2F). They were ellipsoid in shape, smooth, and possessed thin walls. The measured dimensions of these conidia were (5.5) 6.0–8.0 (8.5) × (4.5) 5.0–6.5 (7.0) μm, with a *Q*-value of approximately 1.22. 

Two types of hyphae were observed within the hymenium (Figure 3A). The hyphae of *N. aurantialba* appeared as slender, tube-like structures with a diameter ranging from 2 to 4 μm (Figure 3A–C). Additionally, other hyphae, approximately 3–6 μm in diameter and more robust, and tubular compared to *N. aurantialba,* were identified as belonging to *S. hirsutum* (Figure 3B,D–F). Clamp connections, distinctive structures observed in the septa (cross-walls) of hyphae during cell division, were noted in this study. Abundant clamp connections were present on the hyphae of both species of *N. aurantialba* and *S. hirsutum*, but their morphologies differed. Specifically, *N. aurantialba* exhibited normal clamp connection structures, such as a bulge forming on one side of the septa without a gap (Figure 3A,B), while *S. hirsutum* displayed clamp connections resembling an arched bridge, with a bridge hold forming on one side or both sides of the septa (Figure 3D–F). Additionally, verticillate clamp connections reported by Boidin [21] and Coates [22] were also often observed on *S. hirsutum* hyphae (Figure 3E,F). These are important micromorphological features for identifying the two fungi within basidiocarps.

### 3.3. Basidiospores of Naematelia aurantialba Can Germinate into Hyphae or Bud into Yeast-like Conidia

Based on observation of the morphology of basidiospores and the reproduction process, two reproductive strategies of basidiospores of *N. aurantialba* were identified:

Basidiospores germinated into hyphae. Under favorable conditions, including a relative humidity over 80%, temperature of 20 °C, and nutrient availability, basidiospores became activated. These spores were present on the surface of the basidiocarps (Figure 4B), resulting in a fruiting body that appeared yellow and white, intermingled. When examined under a dissecting microscope, a white powder resembling stacked layers of ice crystal was observed on the surface of basidiocarps (Figure 4D). The activated basidiospores germinated to form germ tubes (Figure 4E), which were elongated structures emerging from the basidiospores, indicating the initiation of germination. The germ tubes could be observed not only at the apiculate but also at any other location of the basidiospores. They continued to elongate and developed into a network of hyphae (Figure 4F). They grew and extended, and eventually, the fruiting body (basidiocarp) became fully covered with a white filamentous layer (Figure 4C). DNA was extracted from the white power (Figure 4B) and ITS sequence was amplified and sequences. The sequence was deposited under accession number of Genbank PQ084746.1. Phylogenetic analysis confirmed that the white power on the surface of the fruiting body is indeed *N. aurantialba* (Figure 1F).

Basidiospores budded into yeast-like conidia. In addition to germinating into hyphae, *N. aurantialba* underwent a form of asexual reproduction called budding from basidiospores. New conidia emerged as outgrowths from the basidiospores, appearing as small bodies on one side of the basidiospores (Figure 5). These new conidia could eventually detach from the parent basidiospores and develop into independent organisms. This budding process from basidiospores was similar to the budding reproduction of yeast. In this case, the basidiospores acted as the parent cells. 

These conidia continued to reproduce asexually through budding, producing yeast-like conidia (Figure 6A) that were subglobose with smooth, thick walls. These characteristics differed from conidia produced by conidiophores within the hymenium. The measured size was (2.0) 2.5–6.0 (6.5) × (1.5) 2.0–5.0 (5.5) μm, with a *Q*-value of approximately 1.16. 

Fungal reproduction involves various mechanisms [23]. It seems that *N. aurantialba* can propagate through both sexual reproduction by producing basidiospores and asexual reproduction by budding from basidiospores, which contributes to its dispersal, reproduction, and long-term survival Notably, the germination of basidiospores into hyphae was often observed on the surface of the fruiting body. Conversely, when basidiospores were collected, budding into yeast-like conidia was more frequently observed on the PDA medium. 

### 3.4. Dimorphic Switching between Yeast-like Conidia and Pseudohyphae or Hyphae

Yeast-like conidia transition into pseudohyphae or hyphae. Apart from budding reproduction, the yeast-like conidia of *N. aurantialba* exhibited two distinct growth forms, pseudohyphae and hyphae. Pseudohyphal growth involved the formation of multicellular filaments, with elongated cells connected by budding. In contrast, the hyphal form represented typical hyphal structures in filamentous fungi. The yeast-like conidia of *N. aurantialba* were incubated on PDA at three temperatures (20 °C, 25 °C, and 28 °C). Pseudohyphae were observed at 28 °C (Figure 6B), while only budding of yeast-like conidia was observed at 20 °C and 25 °C. This suggested that temperature was the key factor for the morphogenesis of pseudohyphae. After trying various culture media and conditions, a transition from yeast-like conidia into hyphae was observed on IDM at 20 °C, 25 °C and 28 °C. (Figure 6C). After being incubated on IDM at 25 °C for a period of 3 days, approximately 20% of yeast-like conidia transitioned into filamentous cells (Figure 6C).

Hyphae switch into conidia. The hyphae-to-conidia transition represents a key developmental process in fungi, allowing them to adapt to changing environmental conditions and establish new colonies. The hyphae, which germinated from yeast-like conidia, consistently elongated, and produced new conidia at the lateral edges of the hyphae (Figure 6D), at their tips (Figure 6E), or both on the side and tips simultaneously (Figure 6F). These conidia were measured 3.5 (3.0)–7.5 (8.0) × 2.0 (1.5)–4.5 (5.0) μm, with a *Q*-value of approximately 1.75.

### 3.5. Haustorium-like Structure and Morphogenesis

The relationship between the coexisting fungi, *N. aurantialba* and *S. hirsutum* remained unclear. During the observation of the hymenium of Jin Er, haustorium-like structures were abundantly observed (Figure 7A–C), each with a clamp connection at the base. Haustoria are specialized structures that are found in some fungi and parasitic plants. It has been reported that haustoria have specialized structures, such as haustorial mother cells or haustorial branches, which are involved in absorbing nutrients from the host [24,25]. Then, we observed the structure and the developmental process in detail, identifying three distinct stages: (1) Protuberances (small projections) with basal clamp connections were formed (Figure 7A). Basal clamp connections are specialized structures that play a crucial role in the formation and function of haustoria. (2) Subsequently, elongation at the protuberances continued, and branches were produced (Figure 7B). These branches may be involved in nutrient acquisition from the host. (3) At the final stage, haustorium-like structures matured and became fully functional (Figure 7C). However, the infection of these structures in the host cell was not successfully observed; we termed them haustorium-like structures. The formation process of the haustorium-like structures was also observed in aerial mycelia cultured on IDM (Figure 7D–I). This suggested that haustorium formation might be influenced by the specific conditions and nutrient availability in the growth medium.

## 4. Discussion

The biological characteristics of edible fungi form the foundation for high-quality cultivation, which is particularly crucial for the intricate “Jin Er” fungi. In this study, we revealed that Jin Er is composed of two fungal species, *N. aurantialba* and *S. hirsutum.* There were two different kinds of hyphae within the hymenium. Basidiospores of *N. aurantialba* can germinate into hyphae or bud into yeast-like conidia. Apart from budding reproduction, the yeast-like conidia of *N. aurantialba* can transform into pseudohyphae or hyphae under certain conditions. The haustorium-like structures were abundantly observed both within the hymenium and on the aerial mycelia cultured on IDM. The developmental process included the formation of protuberances with basal clamp connections, elongation at the protuberances, the production of branches, and maturation. We believe that addressing these fundamental biological issues will be instrumental in understanding the interaction between these two fungi and advancing fruiting body cultivation.

### 4.1. Heterogeneity of Jin Er Fruiting Body

*S. hirsutum* fruiting bodies were observed both in the vicinity of wild and cultivated fruiting bodies (Figure 1A,C). This phenomenon parallels observations in the closely related species *N. encephala* (synonym *T. encephala*), which is naturally surrounded by the fungus *S. sanguinolentum* [26]. In this study, both the hyphae of *N. aurantialba* and *S. hirsutum* were identified in the Jin Er basidiocarps and the ITS sequences also confirmed the coexistence of these two fungi, which was consistent with the results of Cao et al. [27]. Such heterogeneity of the fruiting body was uncommon in fungi. For instance, Yin Er (*Tremella fuciforms*) also exists in a gelatinous state, and another fungal species, *Annulohypoxylon stygium*, must be co-cultured to cultivate *T. fuciforms* fruiting body. However, the basidiocarps Yin Er are made up of homogeneous tissues formed by the mycelia of *T. fuciformis* [28]. Apart from the difference in homogeneity and heterogeneity of the fruiting bodies, there are several other distinctions between Jin Er and Yin Er. Firstly, both *N. aurantialba* and *S. hirsutum* are classified under Basidiomycota, whereas *T. fuciformis* and *A. stygium* belong to Basidiomycota and Ascomycota, respectively. Secondly, pure *T. fuciformis* hyphae can grow and complete the life cycle to form fruiting bodies on synthetic agar media without the presence of the associated fungus [29], whereas *N. aurantialba* currently cannot do so. Thirdly, *A. stygium* is difficult to form fruiting bodies independently, while the fruiting bodies of *S. hirsutum* can be easily obtained. These differences suggest that there may be varying interactions and relationships between these two fungi.

### 4.2. Germination of Basidiospores 

The majority of basidiospores are known to face challenges in germinating and forming hyphae in species belonging to Tremellomycetes [30]. Instead, they primarily reproduce predominantly in a yeast-like form [31,32,33,34,35]. When cultured on PDA medium, *N. aurantialba* basidiospores, collected through the spore ejection method, were observed to produce offspring through budding reproduction (Figure 5), similar to *T. fuciformis*. However, the yeast-like conidia of *T. fuciformis* exhibited mycelial formation when cultured in a liquid medium supplemented with the extracelluar supernatant of *Hypoxylon xianghui* (≡*A. stygium*) [36]. Despite numerous attempts using various media and cultural conditions, we found that the germination of basidiospores to form hyphae was only observed on the basidiocarp surface (Figure 4). This suggested that the basidiospores of the *N. aurantialba* can develop into hyphae under specific environmental conditions. 

### 4.3. Yeast-like Conidia, Pseudohyphal, and Hyphal Morphogenesis of Naematelia aurantialba 

Three different cell types of *N. aurantialba* have been observed: yeast-like conidia, pseudohyphae, and hyphae (Figure 6). The ability of fungi to switch between multicellular hyphae formed from the two compatible mating haploid cells and saprophytic unicellular yeast form is a tightly regulated process known as dimorphism [2,37,38]. Early-diverging basidiomycete lineages Pucciniomycotina [39], Ustilaginomycotina [40], and Tremellomycetes [41] comprise many dimorphic fungi [42], where their morphology varies under different environmental pressures, such as temperature, pH, and nutrition sources (including nitrogen and carbon sources). For instance, thermally dimorphic fungi are triggered by the change in temperature (from 25 °C to 37 °C) to switch to a new form (mycelia–yeast), such as *Sporothrix schenckii* [43], *Penicillium marneffei* [44], and *Histoplasma capsulatum* [45]. The morphological transition of *Pichia pastoris* yeast to hyphae is a response to the extracellular pH (strong acidification) [46]. *Schizosaccharomyces japonicus* undergoes the dimorphic switch from yeast form to hyphal form in response to nitrogen starvation and DNA damage stresses [47,48]. Glucose starvation or low concentration of glucose (0.01%) induces the transition of *Cryptococcus neoformans* [49] and *Candida albicans* [50,51], respectively. In this study, the yeast-like conidia transitioned into pseudohyphae due to a change in temperature from 20 °C to 28 °C (Figure 6B) and switched to filamentous (hyphal) forms on the IDM (Figure 6C). The yeast form typically facilitates dissemination and survival in host environments [43,52], while the filamentous form is associated with nutrient acquisition [46], colonization, and invasion of host tissue, leading to disruption [53]. The filamentous cells of *N. aurantialba* should be important for fruiting body development; however, the roles of the yeast form and the dimorphism remained unsolved.

### 4.4. The Interaction between the N. aurantialba and S. hirsutum 

Fungi growing with or on other fungi have varied relationships, such as saprotrophic, symbiotic, mycoparasitic [54], and even neutral relationships [55,56]. The presence of haustoria is often recognized as a characteristic feature of the mycoparasitic lifestyle. In various instances, such as with the mycoparasitic fungus *Biatoropsis usnearum*, the haustoria produced by its hyphae attach to the cell walls of the lichen-type fungus *Usnea rigida*, showcasing this specific interaction [57]. The haustorial morphology was described in *Tremella* from China [1]. A significant abundance of haustoria has been observed in species like *N. encephala* [34], *T. mesenterica* [58], and *T. mycophaga* [59]. These species exhibit a parasitic relationship with their respective hosts: *S. sanguinolentum*, *Peniophora laeta*, and *Aleurodiscus amorphous*. Within these interactions, haustoria effectively penetrate the cell walls of their host organisms, leading to the creation of distinct micropores at the interface of interaction [32]. In this study, the haustorium-like structures formed by *N. aurantialba* were observed both in the basidiocarps and the mycelia cultured on the IDM (Figure 7); however, further observation is needed to understand the structure and process of haustoria penetrating the host cell. It is worth noting that haustoria have also been identified within the hymenium of *T. macrobasidiata* and *T. variae*, without clear invasion of any other cell [44]. Jin Er mushroom consists of the mycelia of two fungi, *N. aurantialba* and *S. hirsutum*. In the wild, the fruiting bodies of *S. hirsutum* are typically found around the Jin Er fruiting bodies, while during cultivation, the fruiting bodies of *S. hirsutum* only appear around the Jin Er fruiting bodies when conditions are unfavorable. Furthermore, only the mycelia of *S. hirsutum* were observed in the Jin Er mushroom, but no conidia and basidiospores were observed. Upon observing the life cycle of *N. aurantialba*, it was established that *S. hirsutum* is indispensable throughout the entire growth cycle of *N. aurantialba*. Previous research reported *S. hirsutum* absorbed nutrition and transported to fruiting bodies of *N. aurantialba* [35]. The mycoparasitism between *N. aurantialba* and *S. hirsutum* still needs more evidence. 

The production of carpophoroids has been thought to be the result of an *Armillaria* species parasitizing *Entoloma abortivum* sporocarps [60]. However, subsequent studies suggest the opposite: the formation of carpophoroids is the result of *E. abortivum* disrupting the development of *Armillaria* sporocarps [54,61]. In this study, it was observed that the fungus *S. hirsutum* is capable of independently forming fruiting bodies, whereas the pure mycelia of *N. aurantialba* cannot form fruiting bodies on their own. One possible hypothesis is that *N. aurantialba* parasitizes *S. hirsutum*, disrupting its growth and development, ultimately leading to the formation of Jin Er similar to carpophoroids, which consist of *N. aurantialba* and *S. hirsutum* mycelia. In such a scenario, both fungi may benefit from the partnership in some way. For instance, *S. hirsutum* could assist *N. aurantialba* in nutrient uptake, contributing to its growth and development, while *N. aurantialba* may reciprocate by providing resources or a favorable environment that promotes the thriving of *S. hirsutum*. Another intriguing possibility is that *S. hirsutum* might play a regulatory role in the life cycle of *N. aurantialba*. This regulation could occur through chemical signaling or physiological interactions, influencing growth pattern, reproduction, or response to environmental conditions of *N. aurantialba*. Such regulatory mechanisms are common in commensalism relationships [62,63].

## 5. Conclusions

The fact that *S. hirsutum* is essential for the growth and development of *N. aurantialba* and the enigmatic coexistence of *N. aurantialba* and *S. hirsutum* in Jin Er highlight the intricacies of fungal interactions. The yeast-like conidia transitioned into pseudohyphae or filamentous forms, exhibiting dimorphism. Haustorium-like structures were abundantly observed both within the hymenium and in the aerial mycelia cultured on the IDM. However, we failed to observe the infection of these structures into the host cell. Understanding the nature of this relationship could shed light on the ecological significance and functional attributes of both fungi within their natural habitat and advance the cultivation of Jin Er fruiting bodies. 

## Figures and Tables

**Figure 1 jof-10-00557-f001:**
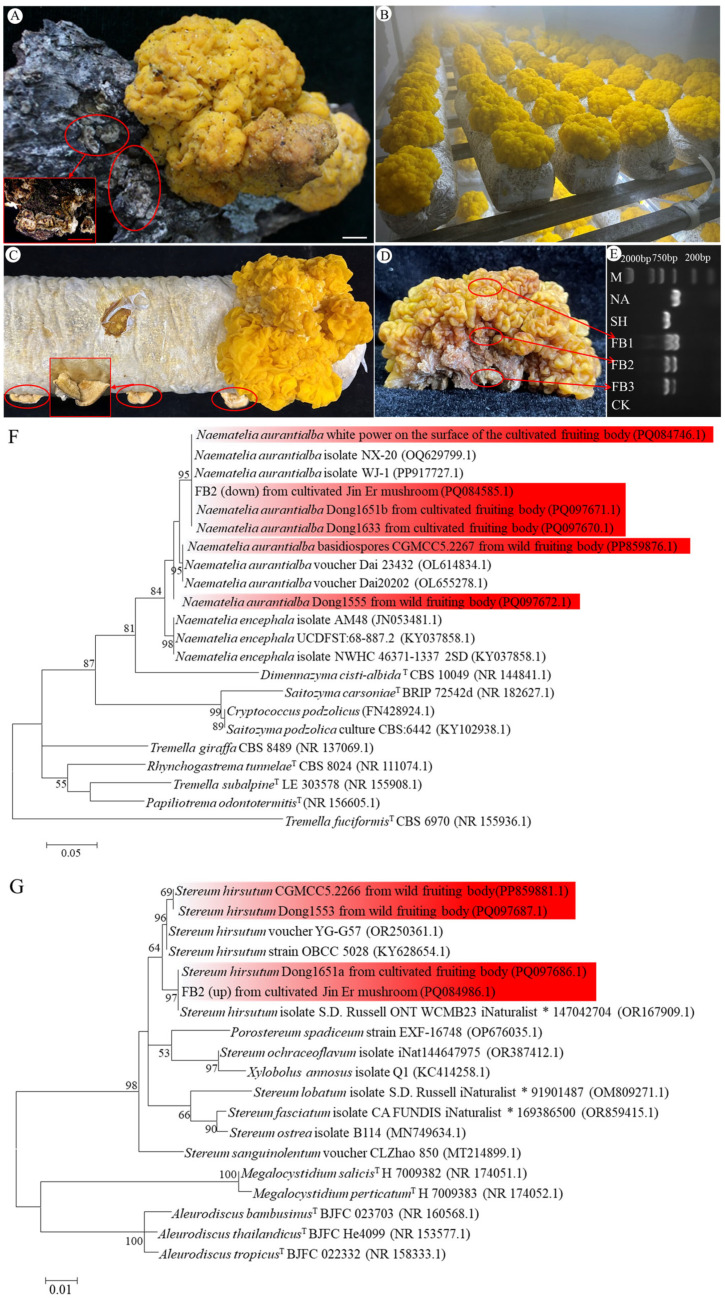
Jin Er fruiting bodies are composed of two fungal species. (**A**) Wild Jin Er fruiting bodies. *S. hirsutum* fruiting bodies adjacent to Jin Er are circled in red; (**B**) artificially cultivated Jin Er to scale; (**C**) artificially cultivated Jin Er and *S. hirsutum* on the growing bag (circled in red); (**D**) artificially cultivated Jin Er fruiting bodies; (**E**) electrophoretic bands of ITS amplification of DNA from different locations of Jin Er. (**F**) Phylogenetic tree based on the ITS sequences of *N. aurantialba* and related species; (**G**) phylogenetic tree based on the ITS sequences of *S. hirsutum* and related species. Sequences in red are from this study. Red scale bar = 1000 μm, white scale bar = 1 cm. M: marker 2 kb; NA: *N. aurantialba* basidiospores; SH: *S. hirsutum* pure mycelia; FB1, FB2 and FB3: top, middle, and bottom of Jin Er fruiting body circled in red; CK: negative control. FB2 (top): upper band of sample FB2 in (**E**); FB2 (bottom): lower band of sample FB2 in (**E**). Superior characters * and T: strain number and type species, respectively.

**Figure 2 jof-10-00557-f002:**
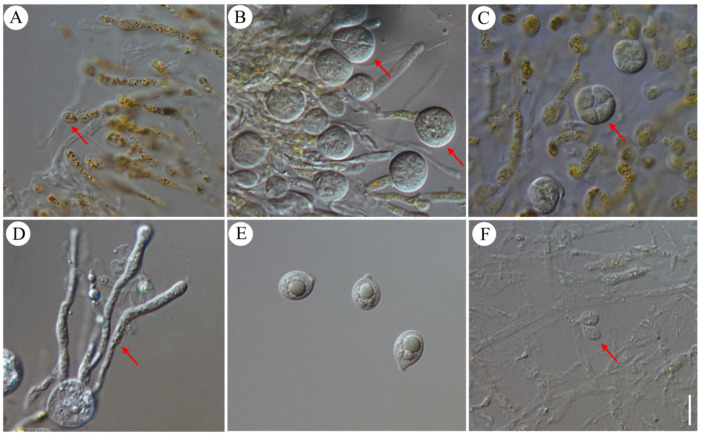
Microscopic structure of Jin Er hymenium. (**A**) probasidia shown by red arrow; (**B**,**C**) mature basidia shown by red arrows; (**D**) wpibasidia shown by red arrow; (**E**): basidiospores; (**F**) conidia are indicated by red arrows. Scale bar: 10 μm.

**Figure 3 jof-10-00557-f003:**
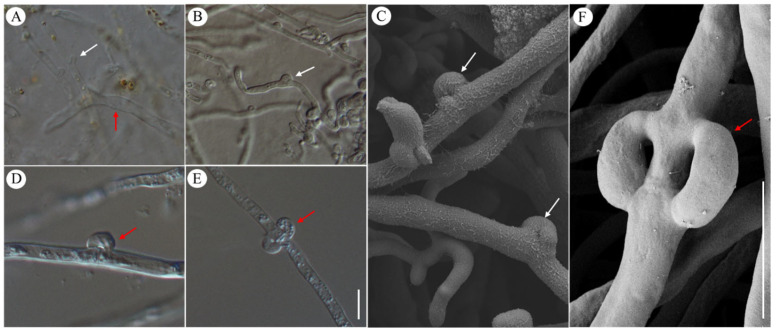
The differences in hyphal morphology between *Naematelia aurantialba* and *Stereum hirsutum*. (**A**) *N. aurantialba* and *S. hirsutum* hyphae in hymenium shown by white and red arrow, respectively; (**B**,**C**) Clamp connections on *N. aurantialba* hyphae shown by white arrows; (**D**–**F**) Clamp connections on *S. hirsutum* hyphae shown by red arrows. Scale bars: 10 μm.

**Figure 4 jof-10-00557-f004:**
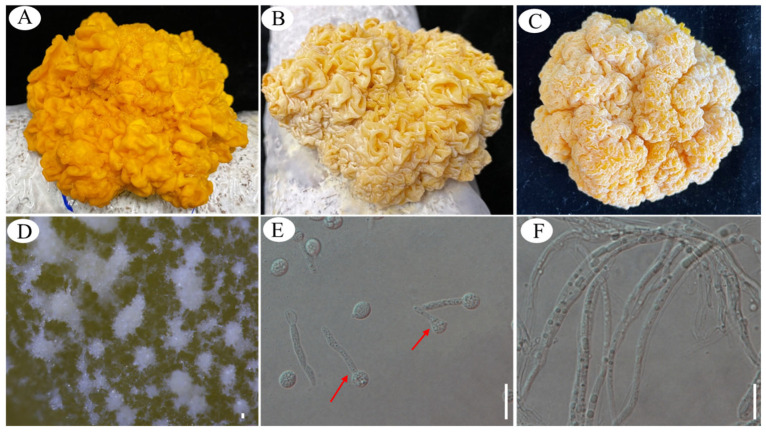
Germ tubes and hyphae generated from basidiospores on the surface of Jin Er. (**A**) Fruiting body with a clean surface; (**B**) fruiting body appearing yellow and white, intermingled; (**C**) surface of basidiocarps covered with white filaments; (**D**) surface of basidiocarps under a dissecting microscope; (**E**) germ tubes formed by basidiospores, shown by red arrows; (**F**) hyphae covering the surface of basidiocarps. Scale bar: 20 μm.

**Figure 5 jof-10-00557-f005:**
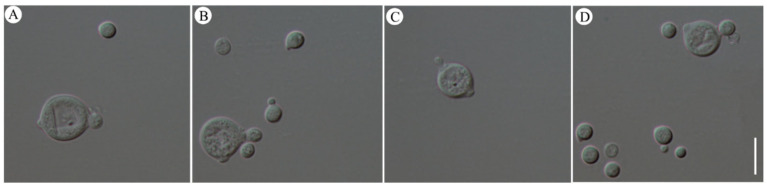
Yeast-like blastoconidia formed by budding reproduction from basidiospores of *Naematelia aurantialba*. (**A**,**B**) Bud formation; (**C**,**D**) abscission of buds. Scale bar: 10 μm.

**Figure 6 jof-10-00557-f006:**
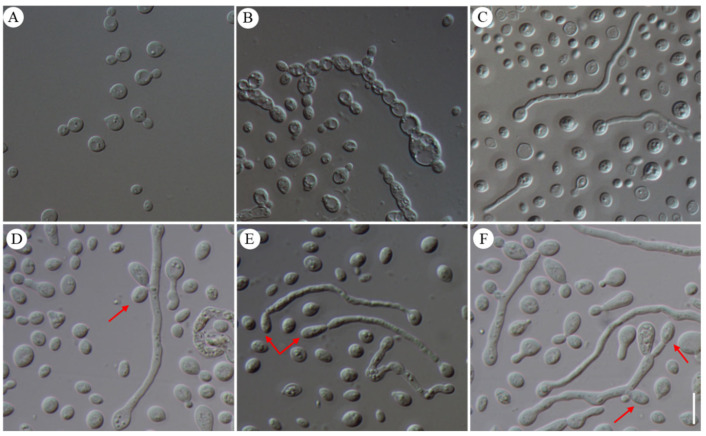
Dimorphic transition between yeast-like conidia and pseudohyphae or hyphae and conidia of *Naematelia aurantialba* produced by hyphae. (**A**) Conidia in bud; (**B**) pseudohyphae transformed from yeast-like conidia on PDA at 28 °C; (**C**) filamentous cells formed by yeast-like conidia on IDM at 25 °C; (**D**–**F**) conidia formed by filamentous cells shown by red arrows. Scale bar: 10 μm.

**Figure 7 jof-10-00557-f007:**
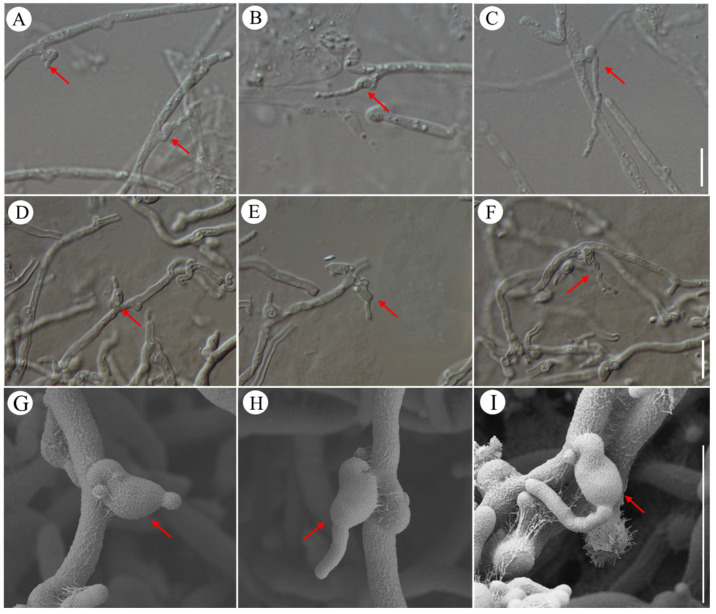
The morphology and developmental process of haustorium-like structures in Jin Er hymenium and mycelia grown from yeast-like conidia. (**A**–**C**) Haustorium-like structures on mycelia in Jin Er hymenium; (**D**–**F**) haustorium-like structures on mycelia grown from yeast-like conidia on IDM, as observed under an optical microscope; (**G**–**I**) haustorium-like structures on mycelia grown from yeast-like conidia on IDM using SEM. Red arrows show haustorium-like structures. Scale bar: 10 μm.

## Data Availability

The original contributions presented in the study are included in the article, further inquiries can be directed to the corresponding authors.

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
