# Peer review of "Fruiting Body Heterogeneity, Dimorphism and Haustorium-like Structure of Naematelia aurantialba (Jin Er Mushroom)"

_jof, 2024, doi:10.3390/jof10080557_

Round 1
Reviewer 1 Report
This article investigates different morphological stages of Naematelia aurantialba, known as Jin Er mushroom in China. The authors utilized morphological and molecular data to show the fruiting body heterogeneity of Jin Er mushroom, which is composed of Naematelia aurantialba and Stereum hirsutum. The authors then investigated the morphology of basidiospore, fungal mycelia, and yeast-like conidia of N. aurantialba. The authors also demonstrated the fungal dimorphism of N. aurantialba based on its culture on varying environmental conditions. Finally, the authors found the haustorium-like structures both in the basidiocarp and in IDM culture. The authors discussed the life strategy of N. aurantialba, its life cycle, and its potential interaction with S. hirsutum in Jin Er mushroom development.
I found this manuscript interesting, especially in the part of fruiting body heterogeneity. However, I think there are several points that do not have strong supporting data/evidence for the authors’ findings. The molecular study of fruiting body heterogeneity is quite minimal and leaves several missing pieces of data that can be used to exclude other possibilities. Some morphological characters, like haustorium-like structures, need to be compared with closely related species. Additional experiments should have been conducted to strengthen the authors’ findings. Please see below for some points that should be addressed.
- I think the manuscript should have been clarified in detail on how they found fruiting body heterogeneity in Jin Er mushroom. Did the authors conduct all observations based on a single fruiting body, or collect multiple ones from multiple sites? If it is based on a single fruiting body, all results may not be rigid enough to generalize the authors’ findings. If the authors’ findings are based on observations from multiple fruiting bodies, then the authors should explain and clarify in the manuscript. The authors can start by saying that the authors observed several Jin Er fruiting bodies from natural habitats (or observed artificially cultivated Jin Er mushrooms). Then, the authors performed molecular and morphological analyses across multiple samples and found the same thing. The authors should also specify the sample used in this study (specimens should be vouchered in a herbarium). Any culture strains isolated from this study should be deposited in the culture repository such as CBS collection or the one in China. ITS sequences generated from this study should be deposited in a public database like GenBank. This way other researchers in the future can trace back the data for reproducibility and referencing.
- My biggest concern after reviewing this manuscript is the molecular study. The author minimally described their materials and methods. How many fruiting bodies did the author perform molecular analyses to test fruiting body heterogeneity? How many trials/locations/replicates per each fruiting body did the authors test? Did the authors have proper negative and positive controls? How did the authors perform molecular identification? Perform a blast search? Did the authors just pick the best hit and assign species identity? Did the authors compare with the sequence from the type strains/specimens of Naematelia aurantialba and Stereum hirsutum? Again, these sequences need to be deposited in the public database.
- Molecular identification is also beneficial to prove other findings reported in this manuscript. For example, the authors said that the basidiospores of Naematelia aurantialba can directly germinate and grow on a fruiting body surface as a ‘white powdery mass’. How did the authors ensure that the white mycelial mat is from N. aurantialba, not from S. hirsutum or other random contaminants? The authors could isolate the white mycelial mat as pure culture, and then do molecular analyses to check species identity. Another example, how did the authors ensure that basidiospores from the Jin Er mushroom are only for N. aurantialba but not S. hirsutum? The authors could collect basidiospore suspension, extract DNA, and run ITS amplification-sequencing to check species identity. This way can be used to support the authors’ findings and exclude other possibilities.
- The authors mentioned that the fruiting body development of N. aurantialba requires the coexistence of S. hirsutum mycelia. However, the manuscript shows little effort for this statement. As there is no previous scientific study (i.e., no reference in the introduction) in this issue, the authors should have conducted additional experiments to demonstrate that the fruiting body development of N. aurantialba depends on the presence of S. hirsutum. Examples are such as cultivation experiments with/without S. hirsutum to see if Jin Er mushrooms can form.
- Other morphology of S. hirsutum in Jin Er mushroom is poorly mentioned. Does S. hirsutum produce only hyphae with clamp connection? Does it produce any conidia or basidiospores? If so, how are they distinguished from the spores from N. aurantialba?
- Did the authors make any efforts to see functional haustoria? For instance, the authors could try co-culturing the mycelium of N. aurantialba and S. hirsutum on an IDM agar plate to see if N. aurantialba form haustoria or haustorium-like structure on S. hirsutum hyphae.
- The authors should have quantification data for dimorphic studies. Since the dimorphic switching is not an all-or-none response, providing percentage of filamentous cells would really help readers to understand the strength of culturing conditions for N. aurantialba growth forms.
Line 17 – 18: Please make a sentence clearer. ‘the homogeneity’ of what? I suggest the sentence as “This heterogeneity of fruiting bodies is indeed distinct from the homogeneous hypha of Tremella fuciformis found in Yin Er mushroom, although its development also requires the presence of another fungus Annulohypoxylon stygium.
Line 51ff: Any backup references here?
Line 56ff: Any previous research, or reference to support this statement?
Line 84: Please provide a proper citation for the ITS1/ITS4 primers
Line 100: The authors should clarify what ‘the white powder’ means. The surface of which part? Hymenium? Or mycelial mat?
Line 103ff : If the author plans to report their morphological observations, I think the authors should mention the following points in Materials and Methods. How many fruiting bodies did the authors collect for morphological studies? Were they from the same location, habitat, and substrate type? If the authors collected the mushrooms from the wild, how did they identify they are ‘Jin Er’? By using a mycological guide, running a dichotomous key, or comparing them with the original description of N. aurantialba? If the authors have the remaining samples, I recommend the authors to deposit them as herbarium vouchered specimens for references in the future. If the authors report the morphology based on artificially grown Jin Er, please specify and clarify in the materials and methods section.
Line 111 – 113: How many locations did the authors do DNA sampling? From how many fruiting bodies? The sampling effort was minimally described in the materials and methods section.
Line 119 – 120: Were the two sequences correctly identified as N. aurantialba, S. hirsutum? Incorrect species assignment is quite common for sequences in GenBank. The authors can compare their sequences with the RefSeq database, which is a collection of DNA data from type strains/specimen, so the authors can ensure the species identity. Did the authors double-checked the identity of sequences retrieved from two-band samples and one-band samples to ensure that the fruiting body is composed of N. aurantialba and S. hirsutum? If so, please clarify.
Line 121 – 122: How did the authors ensure this is a ‘coexistence of mycelia within the basidiocarp’ based on only PCR-sequencing data? It can be other possibilities, like contamination of S. hirsutum spores on N. aurantialba fruiting bodies. Did the authors try to perform PCR-sequencing by using DNA samples extracted from collected basidiospores? If it is just a mycelial coexistence, then the PCR from DNA samples from spore suspension should give one band with a sequence identity for N. aurantialba.
Line 134 – 136: Any size measurements for these structures?
Line 178ff: How did the authors ensure that the white mycelia are from N. aurantialba, not from S. hirsutum or other hyperparasites? Did the authors compare mycelial morphology among N. aurantialba hyphae, S. hirsutum hyphae and white materials. Another easy way to prove it is to isolate the white hyphae, then do DNA extraction-PCR-sequencing and check species identity if it is N. aurantialba.
Line 193ff: Did the authors check the identity of yeasts to ensure it is N. aurantialba? All yeast cells in Figure 5 are globose, while the yeast cells in Figure 6 are a mixture of globose, subglobose, and ovoid cells.
Line 234ff: Please be cautious on the identification of pseudohyphae and true hyphae. The true hyphae typically have a smaller diameter than yeast cells and pseudohyphae. The true hyphae also do not have constriction along a slender filament. To me, arrows in Figure 6C indicate buds from pseudohyphae, not conidia. Even the arrows from panel E are likely to be buds. The one from panel D can be conidia. The authors may use the term ‘blastoconidia’ though.
Line 285 – 287: The sentence is confusing. Did the authors mean that Annulohypoxylon stygium needs to be co-cultured to make Trmella fuciformes produce fruiting bodies? If so, please clarify.
Line 294: Did the authors conduct any experimental trials to see that N. aurantialba cannot produce fruiting bodies when cultured alone?
Line 308 – 309: Did the authors conduct any experimental trials to see if they are chemical or structural composition on the basidiocarp surface that can trigger germination of N. aurantialba basidiospores? One way is to make a crude extract from the fruiting body and treat it on basidiospores to see if they can directly germinate to germ tube.
Line 357 – 359: The authors did not explicitly show their experimental data that the authors went on a trial but not success.
Figure 1: I think it is very hard to see from eyeballs that the red circles in the panel A are S. hirsutum fruiting bodies. The red circles in the panel C are okay despite low resolution. It would be good to indicate a scale bar to reference the fruiting body size. Please indicate fragment lengths of the DNA marker in the panel E.
Figure 2: Please use arrows or arrowheads to guide readers for structures the authors mentioned in the figure legend. For example, the panel C is a mixture of basidium and basidiospores. Are all panels shown with the same magnification? Do the authors have the microscopy image for a sectioned hymenium? This way can really convince the readers that Jin Er produced two types of spores on the hymenium.
Figure 3: It would be great if the authors have a photomicrograph from LM and/or SEM that explicitly shows distinct mycelia from two species within a single hymenium.
Figure 4: It would be nice if the authors have photos showing a gradual change of white mycelia covering Jin Er fruiting bodies on the panel A.
Reviewer 2 Report
The manuscript entitled "Fruiting body heterogeneity, dimorphism and haustorium-like structure of Naematelia aurantialba (Jin Er Mushroom)" studies the fungal interactions and the coexistence of N. aurantialba and S. hirsutum in Jin Er is well written, clear and easy to read. I therefore believe that the paper is suitable for publication in JoF.
1. The authors need to make sure that the title describe the article's topic with sufficient precision. The manuscript investigate the relationship and interaction between Stereum hirsutum and Naematelia aurantialba, which is not reflected in the title.
2. It may help if the authors more clearly state the aims and objectives of the study.
3. The authors can describe in more detail in the Introduction the cultivation of N. aurantialba fruiting bodies requires the presence of S. hirsutum.
4. The authors need to provide information about how the ITS sequences were amplified with 83 ITS1/ITS4 primers.
Reviewer 3 Report
Please see the attachment with detailed comments.
Please see the attachment.

Round 2
Reviewer 1 Report
Comments for authors
I would like to thank the authors for considering my comments to improve their manuscript. I am pretty satisfied with most responses the authors provided, and I think this manuscript is worth for publishing in JoF. Please see below for additional comments.
- Please provide GenBank accessions for other sequencing results used in the analyses. Although the authors see two bands from all sampled regions of Jin Er mushroom, the authors provided sequencing results from only one region. From the phylogenetic trees in Figure S2A and B, there are some sequences that do not have GenBank accessions. So why don’t the authors deposit those sequences too?
- Please provide details for phylogenetic analysis in the Materials and Methods.
- I think Figure S3 can be cited in the introduction, and the authors can refer to it as Figure 1.
- Figure S1A has better representation than Figure 1D – E. I think the authors can replace them the panel S1A to 1D – E.
- Figure S2 is actually a useful piece of information to confirm the fruiting body heterogeneity of Jin Er. The author can make them as other panels in Figure 1. But I would leave this upon the authors’ decision.
- In the author responses, the authors said that “Thirdly, N. aurantialba basidiospores collected by the spore ejection method were confirmed to be pure N. aurantialba by ITS sequence analysis and there is no contamination.” Please mention this in the manuscript too since this is another important information to exclude other possibility. Also, please provide GenBank sequence accession, and include this sequence in the existing phylogenetic analyses.
None.
